# Reliability of blood inflammatory markers at constant real-life workloads over time: Study protocol

Tessy Luger[1]*, Felix Uhlemann[1], Florestan Wagenblast[1], Thomas Läubli[1], Barbara Munz[2,3], Manfred Schmolz[4], Monika A. Rieger[1], Benjamin Steinhilber[1]

**1** Institute of Occupational and Social Medicine and Health Services Research, Eberhard Karls University Tübingen, Tübingen, Germany, **2** Department of Sports Medicine, University Hospital Tübingen, Tübingen, Germany, **3** Interfaculty Research Institute for Sports and Physical Activity, Eberhard Karls University of Tübingen, Tübingen, Germany, **4** HOT Screen GmbH, Reutlingen, Germany

* tessy.luger@med.uni-tuebingen.de

## Abstract

### Background

Work-related musculoskeletal disorders (WMSDs) are prevalent in occupations characterised by high repetition and high force demands. Both factors not only evoke inflammatory and degenerative processes in affected musculoskeletal tissue, but also systemic responses identified by biomarkers in blood serum. Clarifying methodological aspects of biomarkers may provide insights into their predictive role in the pathway of developing WMSDs. This study will primarily assess reliability of systemic inflammatory biomarkers (CRP, TNF-α, IL-6, IL-1β) and immune cell reactivity by repeated measures in workers with constant workloads over time.

### Methods

This observational cross-sectional study will include two groups of workers: exposed group including workers exposed to higher upper-extremity physical workloads, especially affecting the elbow/forearm/hand-area; unexposed group, including office workers exposed to lower upper-extremity physical workloads. Recruited persons are screened against eligibility criteria followed by a medical anamnesis and blood analysis. Enrolled participants undergo nine repeated measurements once every two weeks, taking blood among others. Blood analyses will determine values of systemic inflammatory biomarkers and reactivity of immune cells. The absolute test-retest reliability of biomarkers and immune cell reactivity over time is assessed by the intra-class correlation coefficient applying two-way mixed-effects models. The relative test-retest reliability is assessed by the standard error of measurement.

**Data availability statement:** No datasets were generated or analysed during the current study. For publication of the results of the here outlined study, we will make sure that datasets generated and analysed will be included as supplementary material, available from the corresponding author on reasonable request or available in a repository supported by the Eberhard Karls University of Tübingen (Germany).

**Funding:** The study is financed by the internal research funding program Applied Clinical Research (AKF; No. 465-0-0) of the Medical Faculty of the University of Tübingen (Germany), and own financial resources of the Institute of Occupational and Social Medicine and Health Services Research (IASV). The Institute of Occupational and Social Medicine and Health Services Research receives an unrestricted grant of the employers' association of the metal and electrical industry Baden-Württemberg (Südwestmetall, Germany). This funding sources had no role in the design of this study and will not have any role during its execution, analyses, interpretation of the data, or decision to submit results.

**Competing interests:** Author MS declares to have a financial competing interest, since he is the CEO and CSO of HOT Screen GmbH (Aspenhaustraße 25, D-72770 Reutlingen, Germany). The company HOT Screen GmbH did not play a role in the study design, data collection and analysis plan, decision to publish, or preparation of this manuscript. The remaining authors TLu, FU, FW, TLa, BM, MAR and BS declare that they have no competing interests.

**Abbreviations:** AKF, Applied Clinical Research [Angewandte Klinische Forschung]; BMI, Body Mass Index; CEO, Chief Executive Officer; CLIA, Clinical Laboratory Improvement Amendments; CMDQ, Cornell Musculoskeletal Discomfort Questionnaire; CRP, C-Reactive Protein; CSO, Chief Strategy Officer; CTGF, Connective Tissue Growth Factor; DGUV, German Social Accident Insurance [Deutsche Gesetzliche Unfallversicherung]; DRKS, German Clinical Trials Register [Deutsches Register Klinischer Studien]; E. coli, Escherichia coli; ELISA, Enzyme-Linked ImmunoSorbent Assay; GGT, Gamma-Glutamyl-Transferase; HbA1c, Hemoglobin A1C; HDL, High-Density Lipoprotein

## Discussion

Knowledge of and models currently describing the pathological role of systemic inflammatory biomarkers are based on highly-controlled laboratory rat experiments. This study has the strength of assessing a human population under real-life conditions. The major challenge is in participant recruitment given the intensive and complex study design. The results of this study could provide fundamentals for initiating a cohort study and be used for developing work-related stress-recovery concepts for occupations with different physical demands to identify workers who may be at risk for developing WMSDs. German Clinical Trials Register (DRKS00031872, 25 May 2023).

## Introduction

Musculoskeletal disorders (MSDs) are one of the most common causes of work absenteeism [1]. The majority of MSDs in the working force are judged to be work-related, meaning that workload itself is considered an important cause in a multi-causal disease process [2]. Important stress factors are high force demands, repetitive activities, and awkward body postures, which are required during a substantial part of working hours [3,4]. Health insurance statistics showed that in occupational groups whose activities are characterized by high force demands and repetition, there is a more than fourfold increase in the risk of, for example, tendinitis/tenosynovitis of the elbow [2].

Several workload models have tried to visualise the pathways of work-related factors eventually leading to work-related MSDs (WMSDs), all of them based on the so-called stress-strain concept by Rohmert [5]. Here, *stress* stands for the objective work characteristics and *strain* for all physiological and psychological reactions within the workers. A central principal in ergonomic work design is that the work requirements should not lead to a *strain* that exceeds *long-term performance*. Occupational health and safety enforcement authorities worldwide use standardized assessment procedures to prevent WMSDs. However, the preventive effect of such work design measures is insufficient, and the question arises as to whether previously ignored stress factors are significant for the occurrence of WMSDs [6,7].

In this respect, we postulate that too little attention has been paid to recovery in the work physiology and occupational medicine literature, which may actually be of particular importance for the chronicity of WMSDs, especially in the modern workplace [8]. A rat model, developed by Barr, Barbe and other colleagues [9,10], showed that highly repetitive voluntary movements alone but also while interacting with force demands elicit inflammatory and degenerative processes in directly affected muscle or muscles, tendons, nerves and bones and are associated with systemic responses and neurological reorganization at the levels of the spinal cord and brain. Systemic serum inflammatory indicators that inversely correlated with decreased grip strength in rat models [11] were the cytokines IL-6, TGFB1, and TNF-α and the collagen turnover proteinase MMP2. Gao, Fisher [11] concluded that serum IL-6,

cholesterol; HPLC, High-Performance Liquid Chromatography; IASV, Institute of Occupational and Social Medicine and Health Services Research; [Institut für Arbeitsmedizin, Sozialmedizin und Versorgungsforschung]; ICC, Intra-class Correlation Coefficient; IL-1β, Interleukin-1 beta; IL-6, Interleukin-6; LDL, Low-Density Lipoprotein cholesterol; LPS, LipoPolySaccharide; MHC, Major Histocompatibility Complex; miRNA, Micro Ribonucleic Acid; MMP2, Matrix Metalloproteinase-2; MSD, Musculoskeletal Disorders; NRS, Numeric Rating Scala; PPT, Pressure Pain Threshold; RPP, Rating of Perceived Pain; SEM, Standard Error of Measurement; SPIRIT, Standard Protocol Items: Recommendations for Interventional Trials; TGFB1, Transforming Growth Factor Beta-1; TLV for HAL, Threshold Limit Values for Hand Activity Level; TNF-α, Tumor Necrosis Factor Alpha; TStim, T cell activation; WMSD, Work-related MusculoSkeletal Disorder.

TGFB1, TNF-α, MMP2 but also the glycoprotein CTGF may serve as biomarkers of WMSDs, although studies in humans are needed. With regard to acute inflammatory responses in muscles and tendons caused by physical stress, best-studied biomarkers are IL-1β, TNF-α, and especially IL-6, whose concentration in serum can increase many times over shortly after exposure [12]. In addition, the acute phase protein CRP is frequently determined as routine laboratory parameter, whose concentration also responds immediately following physical exposure [13] as well as in the longer term following physical exposure [14]. Assuming that acute inflammatory processes are important for regeneration and adaptation of skeletal muscle [15], chronic inflammation leading to fibrosis may develop instead of recovery and strengthening muscle and tendon tissues, for example in cases of work-related exposures being disrupted by subsequent workloads [16]. These assumed pathway from work-related exposure to the development of WMSDs is visualised in a model presented in Fig 1.

Investigating the suggested biomarkers in a human population and clarifying open methodological aspects may provide insights into the predictive role of these biomarkers in the pathway of developing WMSDs. The current pilot study could provide fundamentals for initiating a future epidemiological cohort study. The preliminary results of the current study and (if applicable) of the future cohort study, selected biomarkers could be used in the long term to develop meaningful stress recovery concepts and related occupational health check-ups for occupations with different physical demands, thereby detecting individual workers suffering from work-related overstrain early and consequently reducing the risk of WMSDs.

**Aims and objectives**

This study is to be carried out in order to clarify open methodological aspects for the subsequent implementation of a long-term study (cohort of occupationally exposed workers). The primary objective of this pilot study is:

(1) To assess the *reliability* of *systemic biomarkers of inflammation* (i.e., CRP, TNF-α, IL-6, IL-1β) and the *reactivity of immune cells* by repeated measures in workers with constant workloads *over time*.

In addition, we have formulated several secondary objectives to be investigated exploratory:

(2) To assess the *reliability* of (a) *rating of perceived pain*, (b) *pressure pain thresholds* and (c) *functionally triggered pain* by repeated measures in workers with constant workloads *over time*.

(3) To assess the *correlation* between *systemic biomarkers of inflammation* and (a) *rating of perceived pain*, (b) *pressure pain thresholds* and (c) *functionally triggered pain*.

(4) To compare the (a) *level of systemic biomarkers of inflammation*, (b) *rating of perceived pain*, (c) *pressure pain thresholds* and (d) *functionally triggered pain* between *two groups* of higher (exposed group) and lower workload (control group).

| | STUDY PERIOD | | | | | | | | | | | |
|---|---|---|---|---|---|---|---|---|---|---|---|---|
| | Enrolment | Allocation | Post-Allocation | | | | | | | | | |
| **TIMEPOINT** | **$-t_1$** | **$t_0$** | **$t_1$** | **$t_2$** | **$t_3$** | **$t_4$** | **$t_5$** | **$t_6$** | **$t_7$** | **$t_8$** | **$t_9$** | |
| **ENROLMENT** | | | | | | | | | | | | |
| Workload assessment | X | | | | | | | | | | | |
| Eligibility screen | X | | | | | | | | | | | |
| Informed consent | X | | | | | | | | | | | |
| Allocation | | X | | | | | | | | | | |
| **ASSESSMENTS** | | | | | | | | | | | | |
| Joint mobility according to neutral-zero-method | X | | X | X | X | X | X | X | X | X | X | |
| Cornell Musculoskeletal Discomfort Questionnaire | X | | X | X | X | X | X | X | X | X | X | |
| 11-point numeric scale for rating of perceived pain | | | X | X | X | X | X | X | X | X | X | |
| Pressure pain thresholds | | | X | X | X | X | X | X | X | X | X | |
| Functionally triggered pain | | | X | X | X | X | X | X | X | X | X | |
| Blood sampling | X | | X | X | X | X | X | X | X | X | X | |

**Fig 1. Model of the relationship between workload, chronic inflammation, and work-related musculoskeletal disorders.** The model is based on the fundamentals Barbe and Barr [16] and Barbe, Gallagher [9].

## Materials and methods

### Study design and study setting

This is an observational cross-sectional study of approximately four months that was registered at the German Clinical Trials Register on 25 May 2023 (Identifier: DRKS00031872). This protocol adheres to the Standard Protocol Items: Recommendations for Interventional Trials (SPIRIT) 2013 Statement [17] and a guided checklist is added as S1 Table [18].

The study takes place in Southern Germany (Federal State of Baden-Württemberg close to the Tübingen District) and ethics approval was granted by the Ethics Committee of the University Hospital of Tübingen and Medical Faculty of the University of Tübingen on 29 March 2023 (Identifier: 125/2023BO2; see S2 File for Ethical Proposal and S3 File for Ethical Approval). Data can be collected either at our work physiology laboratory when it applies to participants who have somewhat flexible working hours and live or work near our work physiology laboratory or at the work location when it applies to participants who do shift work and do not live or work near our work physiology laboratory.

## Target population

This study will include two groups of workers. One group (exposed group) will include workers exposed to higher physical workloads on the upper extremities, especially affecting the elbow/forearm/hand-area. These group of workers will be recruited from industrial companies that focus on manual assembling and mainly deal with shift work. The other group (unexposed group) will include particularly office workers who are exposed to lower physical workloads on the upper extremities often without shift work. Since we currently lack the necessary knowledge to calculate an appropriate sample size for both the exposed and the unexposed groups, we achieve to include sample sizes of 20 participants for both groups, furthermore striving to include half men and half women in both groups.

**Exposed group.** Workers are considered *eligible* for study participation if they meet the following *inclusion criteria*:

i. Aged between 18 and 67 years.

ii. Continuous employment in the current type of work for at least one year at the time of inclusion.

iii. Workplace where physical loads on hands and arms exceed the Action Limit of the Threshold Limit Values for Hand Activity Level [TLV for HAL; 19, 20].

iv. A daily repetitive strain on one or both hands on ≥ 4 days a week for a total of ≥ 20 hours a week.

v. Suffer from varying degrees of musculoskeletal pain in the elbow/forearm/hand-area (including no musculoskeletal pain) as assessed using the Cornell Musculoskeletal Discomfort Questionnaire [CMDQ; 21].

vi. Written informed consent to study participation and data protection.

Workers are considered *ineligible* for study participation if they meet one or more of the following *exclusion criteria*:

i. Suffering from musculoskeletal pain in body regions other than the elbow/forearm/hand-area, which is equal to or exceeds the severity reported for the elbow/forearm/hand-area.

ii. Suffering from chronic inflammatory conditions (e.g., rheumatism or lupus) or metabolic diseases (e.g., metabolic syndrome or hypercholesterinaemia), since these conditions are known to increase inflammatory markers systemically.

iii. Taking regular medications known to interact with inflammatory biomarkers [e.g., statins, ibuprofen; 22].

iv. Status after acute trauma that affects joints or muscles and/or is associated with acute inflammation.

v. Relative (e.g., fear of blood) and absolute (e.g., taking anticoagulants, few suitable veins that should only be used for therapeutic interventions) contraindications for repeated blood sampling.

Note that an occupational physician interpreted the differential blood analyses in the screening process – which apply to some of the above-mentioned eligibility criteria – and advised on study enrollment of the screened persons.

**Unexposed group.** For workers exposed to lower physical loads, the same inclusion criteria apply as for the exposed group, with the exception of: iii) office workplaces should not exceed the TLV for HAL; v) not suffering from any musculoskeletal pain. The exclusion criteria for workers in the unexposed group are also similar to those of the exposed group, with the exception of: i) not suffering from musculoskeletal pain at all.

All study participants will receive an expense allowance of EUR 500 for complete participation and partial compensation for incomplete participation.

## Outcomes

The primary outcomes are the blood-serum levels of IL-6, TNF-α, CRP and IL-1β at the time of blood collection at six to nine repeated time points. Secondary outcomes include the rating of perceived pain (RPP) in the elbow/forearm/

hand-area, pressure pain thresholds and functionally triggered pain during the working week or at the same time-point blood is collected at nine repeated time points (we set the required minimum at six repeated measures). Other outcomes are used descriptively to characterize the study population, including age, sex, smoking status, physical exposure at work (based on TLV for HAL), results of the medical examination including body joint mobility.

## Participant timeline

The overall study is expected to last three calendar years. The first 1.5 years were needed for preparations, including obtaining a positive ethics vote and organizing research staff and materials. Participant recruitment started in 14 May 2024 and the first participants could be included in 15 November 2024. For organizational reasons, i.e., a maximum of two measurements per exposed and unexposed group per measurement day, we had to plan two measurement blocks. The first measurement block started 13 January 2025. Recruiting participants will continue until August 2025 and the second measurement block can only start after the last participant has been included. The measurements, including data collection, are expected to last until December 2025 at the latest.

Participants for the exposed group are recruited through direct contact with occupational physicians, health and safety officers, and company or department managers at various industrial companies with appropriately designed working stations (i.e., probably including assembly lines with shift work). All the different workstations and areas of potential participants for the exposed group are rated with TLV for HAL by two independent raters at the same time by observing at least three cycles of the particular work step. Participants for the unexposed group are recruited directly face-to-face or through advertisement using announcement e-mails, flyers, and pamphlets. The workplaces of potential participants for the unexposed group are also rated with TLV for HAL by two independent raters at the same time by observing at least five consecutive minutes.

If an individual is interested in participating in the study, an interview by phone will be conducted to pre-screen the interested person against the eligibility criteria. Based on this, an appointment is scheduled for a full screening during which the interested person is informed about the study goals and the right to terminate study participation at any time without giving any reason, before providing informed consent and permission to use and publish the collected data in anonymously (for more information, see: **Declarations**).

This brief introduction is followed by an extensive medical anamnesis including assessments of mobility of body joints by the neutral-zero-method [23], collection of demographic data, pressure pain thresholds [24], functionally triggered pain [25], and perceived musculoskeletal pain assessed by the CMDQ [21] are carried out by a researcher and a physician. At the end of the screening, blood samples (~11 ml) are taken by a physician (Fig 2).

After the person has been included for study participation, the nine biweekly appointments are scheduled: three on Mondays, three on Wednesdays, and three on Fridays. The participant is quasi-randomly assigned (balanced) to one of six possible measurement-day sequences. Participants are asked to take their appointments before their regular work shifts. Each of the nine follow-up measurements includes the joint mobility assessment via the neutral-zero-method on the forearm-region [23], perceived musculoskeletal pain assessed by the CMDQ [21], rating of perceived pain at four time-points using an 11-point numeric rating scale (NRS) [26], pressure pain thresholds [24], functionally triggered pain [25], and collection of blood samples (~18 ml).

## Data collection and analysis

**Physical workload assessment.** The workload on the upper limbs is assessed by using two different methods for the exposed group and the unexposed group. The TLV for HAL [19,20] is used at workplaces of participants in the exposed group, assessing the physical load on hands and arms. Only workplaces where this physical load exceeds the Action Limit of the TLV for HAL are considered for the study to ensure that there is a clear discrepancy with the unexposed group. The TLV for HAL showed fair (Cohen κ of 0.34) to strong (Spearman $\rho$ of 0.65) inter-rater reliability [27].

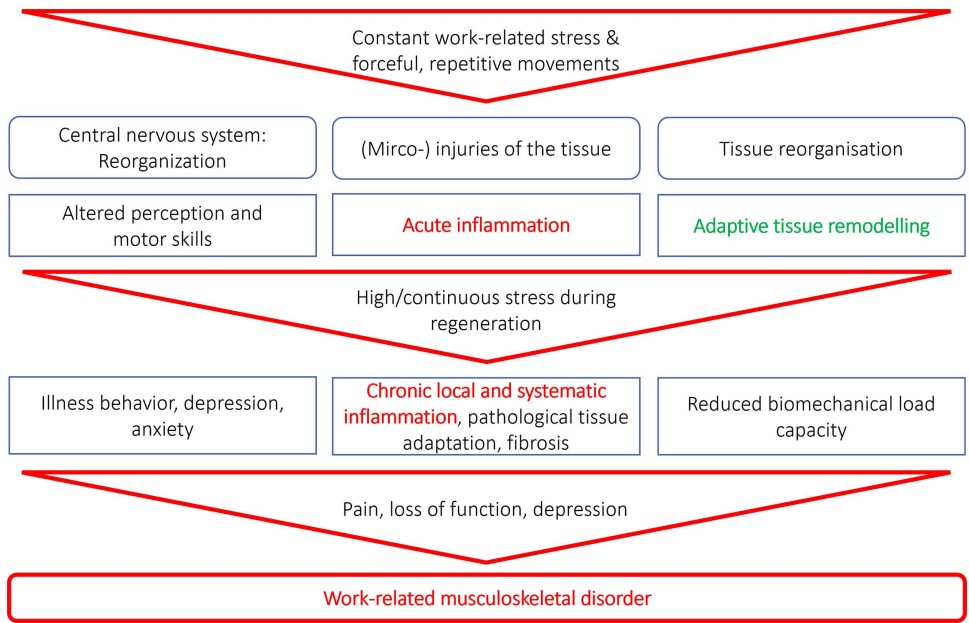

**Fig 2. Schedule of enrolment and assessment, based on the SPIRIT Diagram [17].**

**Medical anamnesis, eligibility criteria and demographic data.** Taking the medical history follows a standardized written interview guide covering health status, general illnesses, suffering from musculoskeletal pain in body regions other than the elbow/forearm/hand-area, suffering from chronic inflammatory or metabolic diseases, acute trauma affecting joints or muscles, taking regular medications that interact with inflammatory biomarkers (e.g., statins, ibuprofen by case discussion) [22], and contraindications for repeated blood sampling (see also exclusion criteria). The specified medical examination is performed according to the guidelines for physicians as provided and developed by the German Social Accident Insurance [28]. The neutral-zero-method [23] is a known method among physicians to assess joint mobility. During the nine repeated measurements, the joint mobility of the hand-arm system only will be assessed using the neutral-zero-method. Specifically, the maximal active range of motion of the following joints is measured using a goniometer: elbow flexion and extension, forearm pronation and supination, wrist dorsal and palmar flexion, and wrist ulnar and radial abduction.

Eligibility criteria are checked by the physician. The demographic data of participants are collected, including: body weight, length, age, gender, leisure time activity level (i.e., sport in the past four weeks in h/week), smoking status, handedness and nationality. Both checklists are conducted in face-to-face interview format, where the physician reads the questions aloud and notes or ticks the answers.

**Musculoskeletal pain.** The German CMDQ is used to map in which body areas (main) musculoskeletal pain is perceived. This questionnaire showed a good validity (Cohen $\kappa$ 0.38–1.00; Spearman $\rho$ 0.40–1.00), a satisfactory internal consistency (Cronbach $\alpha$ 0.75–0.82) and medium to substantial test-retest reliability (Spearman $\rho$ 0.56–0.72) [21]. The CMDQ is conducted in a face-to-face interview format, where the researcher reads the questions aloud and notes or ticks the answers.

The 11-point NRS is used to rate perceived musculoskeletal pain (i.e., RPP) in the elbow-forearm-hand region. The 11-point NRS handles a categorical answer format with endpoints 0 (no pain) and 10 (strongest imaginable pain) and is characterized by a low error rate, high acceptance, easy handling and high sensitivity [29,30]. The NRS-11 has good to excellent validity (Pearson $r$ 0.94 with the 100-mm visual analogue scale) and excellent reliability (ICC 0.95) when

assessing musculoskeletal pain [26]. The RPP is collected at four time-points during each of the nine measurements. The first RPP (i.e., participant's current RPP) is collected immediately before pain provocation (i.e., pressure pain thresholds and functionally triggered pain). On the Friday or Saturday of the work week, the other three values are collected reflecting the past work week (i.e., minimum RPP, maximum RPP and RPP by night), for which the participant is called by one of the researchers involved in the project.

**Blood sampling during screening.** By the end of the screening, blood is collected for performing differential blood and laboratory chemical analyses to verify that participants are free from acute infection or metabolic diseases. Three blood samples are collected through a single venous puncture (Safety-Multifly® needle, 21G x 3/4", green, 120 mm tube length; Sarstedt AG & Co. KG, Nümbrecht, Germany) preferably from the median cubital vein at the bend of the participant's preferred arm: lithium-heparin tube (S-Monovette® LH, 5.5 ml, cap orange, 75 x 15 mm; Sarstedt AG & Co. KG, Nümbrecht, Germany), EDTA-K3E tube (S-Monovette® EDTA K3E, 2.6 ml, cap red, 65 x 13 mm; Sarstedt AG & Co. KG, Nümbrecht, Germany), Fluoride/EDTA tube (S-Monovette® FE, 2.7 ml, cap yellow, 75 x 13 mm; Sarstedt AG & Co. KG, Nümbrecht, Germany). All other standard operating procedures for hygienic blood sampling are adhered to. Various metabolic and inflammatory blood parameters are gathered, including: gamma-glutamyl-transferase (GGT), HbA1c (HPLC and glycated haemoglobin), high- and low-density lipoprotein cholesterol (i.e., HDL and LDL, respectively), triglyceride, enzymatic creatinine and large blood count including thrombocytes. The blood values are weighted against standard values and based on this a decision is made to include or exclude the screened person.

**Pressure pain thresholds.** Pain is provoked using a digital pressure algometer (CE-certificated; ATORN ZD2) on five bilateral anatomical sites of interest to map so-called pressure pain thresholds (PPT), a methodology also used in clinical practice [31]. The PPT procedure showed to have a very high intra-rater reliability [Cronbach α 0.94–0.98; 24]. The five sites are selected based on an extensive literature search, including: (1) carpal tunnel [32], (2) lateral and (3) medial epicondyles [33], (4) extensor carpi radialis brevis and (5) flexor carpi radialis [34]. The measuring tip (1 cm$^2$) of the algometer is placed at a 90-degree-angle to the skin's surface – for exact locations, see Table 1 – while gradually applying increased pressure by ~3 N/s until the participant experiences the slightest pain sensation. Without seeing the pressure value, the participant is instructed to express pain by either saying it or raising their hand of the non-involved arm when the first slight pain sensation is felt. This procedure is repeated on each bilateral anatomical site for three times, after which the average value (N) for each bilateral anatomical site will be used for statistical analyses.

**Functionally triggered pain.** Functionally triggered pain is examined by adopting the protocol of Garg, Hegmann [25], including three examinations. First, the participant executes maximal force during closing of the fist until a pain trigger follows (if any). Second, spontaneous pain is assessed (if any) when moving the hand against resistance in the six directions supination, pronation, lateral flexion radial, lateral flexion ulnar, palmar flexion, and dorsal extension.

**Table 1. Location of the algometer's 1–cm$^2$ measurement tip to determine PPT at the five bilateral anatomical sites.**

| Anatomical site | Location of algometer measurement tip |
|---|---|
| Carpal tunnel | 1 cm distal to the great wrist groove in between the thenar and the hypothenar. |
| Lateral epicondyle | The soft tissue about 1 cm distal from the origin of the extensor carpi radialis brevis on the lateral epicondyle. |
| Medial epicondyle | The soft tissue about 1 cm distal from the origin medial common flexor tendon on the medial epicondyle. |
| Extensor carpi radialis brevis | 3-4 fingers distal from the lateral epicondyle, confirmed by active muscle contraction. |
| Flexor carpi radialis | One-third from the line between the medial epicondyle and the great wrist groove, confirmed via active muscle contraction. |

 

Third, spontaneous pain is assessed (if any) when moving the forearm against resistant in the two directions flexion and extension. The researcher provides sufficient resistance for the participant to develop maximum force for every movement, except for closing the fist. The researcher notes whether the participant experienced pain (yes) or not (no).

**Blood sampling during repeated measures.** Blood samples are collected through a single venous puncture preferably from the median cubital vein at the elbow bend of the participant's preferred arm. All other standard operating procedures for hygienic blood sampling are adhered to. Three different procedures are followed to treat the different blood samples. All laboratory analyses are performed blinded with regard to the group (exposed vs. non-exposed) the study participant's blood belongs to. First, a lithium-heparin tube (S-Monovette® LH, 5.5 ml, cap orange, 75 x 15 mm; Sarstedt AG & Co. KG, Nümbrecht, Germany) is filled and stored at room temperature, after which it is analysed by enzymatic-photometric analysis by the Central Laboratory of the University Hospital of Tübingen (Hoppe Seyler Straße 3, 72076 Tübingen, Germany) to determine CRP and IL-6 values.

Second, TruCulture-Tubes (HOT Screen GmbH, Aspenhaustraße 25, 72770 Reutlingen, Germany) are prepared (S-Monovette® neutral Z, 2.7 ml, cap white, 66 x 11 mm; Sarstedt AG & Co. KG, Nümbrecht, Germany) before study start in batch with a stimulating environment (i.e., for inducing the production of cytokines, chemokines and growth factors) in a volume of 2 ml buffered media, and maintained (i.e., stored) at −20°C at the Institute of Occupational and Social Medicine and Health Services Research of the University Hospital of Tübingen (Wilhelmstraße 27, 72074 Tübingen, Germany) until time of use. The stimulant that was used, included LPS + TStim; LPS is a lipopolysaccharide from the bacterial endotoxin Escherichia coli (E. coli, O55:B5), and TStim is an antibody-based mimic of a superantigen initiating a T cell response by cross linking the T cell receptor and major histocompatibility complex (MHC) on monocytes/ macrophages. The exact time-point (hh:mm) of blood sample collection is noted on a label specifically designed for this study that also contains information on participant's sex, age, date and measurement repetition. Directly after labelling, the TruCulture-Tubes (non-stimulated and stimulated) are filled with 1 ml of whole blood, placed in a dry metal block thermostat (i.e., with heating trays; Type V.69107.61220; VLM GmbH, Bielefeld, Germany pre-heated to 37.0°C) and incubated for exactly 24 h. At the end of the 24-hour incubation period, both the un-stimulated and stimulated tubes are opened to insert a Seraplas® valve filter (white, for separation of the TruCulture supernatants; Sarstedt AG & Co. KG, Nümbrecht, Germany) from the top until 1–2 mm above the sedimented cells to separate them from the different cellular layers of the supernatant and stop the stimulation reaction. With the valve installed, the tubes will be frozen at –20°C until final analysis is scheduled after data collection is completed. An exploratory analysis of the un-stimulated and stimulated TruCulture tubes is performed by HOT Screen GmbH applying Luminex® xMAP® technology (DiaSorin S.p.A., Saluggia, Italy). Since we currently do not have information available regarding the direction in which the target parameters might change, the goal of these exploratory analyses will be to identify patterns related to pathogenesis.

Third, an EDTA-K3 tube (S-Monovette® EDTA K3, 9.0 ml, cap red, 92 x 16 mm; Sarstedt AG & Co. KG, Nümbrecht, Germany) is filled and placed in a water-ice bath to cool to 3–4°C. Within two hours after blood sampling, the tube is centrifuged at 1500G at 4°C for 10 min. Plasma is stored at –70°C in aliquots at the laboratory of the Department of Sports Medicine of the University Hospital of Tübingen (Hoppe-Seylerstraße 3, 72076 Tübingen) until the final analysis is scheduled after the study's data collection is completed. The thawed aliquots with plasma will be analysed for CRP, TNF-α, IL-6, and IL-1β using commercial enzyme-linked immunosorbent assay (ELISA) kits from eBioscience (Thermo Fisher Scientific, San Diego, CA, USA) according to the manufacturer's instructions by the same Department of Sports Medicine. Residual blood plasma will be kept stored at –70°C in aliquots at the laboratory of the Department of Sports Medicine of the University Hospital of Tübingen (Hoppe-Seylerstraße 3, 72076 Tübingen) for possible future use such as the exploratory analysis of further molecular markers, such as miRNA [35].

The blood analyses performed by HOT Screen GmbH (Reutlingen, Germany) and the Department of Sports Medicine of the University Hospital of Tübingen (Germany) will be performed blinded, i.e., the persons performing the laboratory analyses are blinded with regard to the group to which the study participants belong to (exposed vs. unexposed) and with

regard to the repeated measurements. The blood analyses performed by the Central Laboratory of the University Hospital of Tübingen (Germany) will be performed by a group of people who are only blinded to results of previously analysed samples.

## Statistical analysis

For all statistical analyses, we intend to use SPSS (IBM® SPSS® Statistics 28.0, Armonk, NY, USA).

**Primary objective.** For assessing the reliability of systemic biomarkers of inflammation (i.e., CRP, TNF-α, IL-6, IL-1β) over time, we will create Bland-Altman plots for providing visual representations of the difference between two repeated measures and the average of two repeated measures for both the exposed and unexposed group. The Bland-Altman plots will be used to identify any systematic bias in the data. For assessing absolute test-retest reliability, we will apply a two-way mixed-effects model with a single rater for both consistency and agreement to calculate the intra-class correlation coefficient for the exposed and unexposed group [ICC; 36, 37]. For assessing relative test-retest reliability, we will use the standard error of measurement [SEM; 37].

We will also exploratory build different ICCs to obtain information about how many repeated measurements result in a stable ICC. In other words, from how many repeated measurements the ICC is above an acceptable limit.

**Secondary objectives.** We will use the same procedures for determining reliability as described above, only with different parameters as repeated measurements, i.e., rating of perceived pain, pressure pain thresholds and functionally triggered pain.

For assessing the correlation between single systemic biomarkers of inflammation and rating of perceived pain, we will use the Spearman ρ correlation coefficient. For assessing the correlation between single systemic biomarkers of inflammation and pressure pain thresholds, we will use the Pearson $r$ correlation coefficient. For assessing the correlation between single systemic biomarkers of inflammation and functionally triggered pain thresholds, we will use the point-biserial correlation coefficient. For assessing the correlation between the systemic biomarkers of inflammation IL-6 assessed by different laboratory procedures (i.e., enzymatic-photometric analysis versus ELISA), we will use the Pearson $r$ correlation coefficient.

For comparing the (a) level of systemic biomarkers of inflammation, (b) rating of perceived pain, (c) pressure pain thresholds and (d) functionally triggered pain between the two groups of higher and lower physical workload, we will use mixed model analyses.

**Missing values.** In case of missing values, we will analyse only available data, which means that we include the number of repetitions corresponding to the subjects with the smallest _ lowest number of repetitions (i.e., per definition at least six repeated measures; see **Outcomes**), i.e., ignoring missing data. However, ignoring missing values may create a biased estimate, whereas imputing missing values will produce a more unbiased result [38]. Thus, in addition to complete data analysis, we will impute missing data on all data (i.e., biomarkers, pain intensity, pain thresholds, triggered pain) using replacement values according to the best fitting imputation method [39].

## Data management and confidentiality

Physically and digitally collected data of both potential and enrolled participants will be numerically pseudonymised by assigning a randomly generated, two-digit identification number to maintain confidentiality. For publication purposes, any participant-related data will be anonymised. The physically collected data will be stored in locked cabinets at and the digitally collected data will be saved on a separate server of the Institute of Occupational and Social Medicine and Health Services Research (University Hospital of Tübingen, Germany) with limited access. A decoding list together with the physically signed inform consent forms will be stored in separate locked cabinets of the Institute of Occupational and Social Medicine and Health Services Research (University Hospital of Tübingen, Germany) with limited access. Only the

principal investigators of the study have direct permission to access the decoding list and the physically and digitally collected data. All data will be stored for a period of ten years after publication of the study results, after which physical data are destructed in disposal boxes for data protection at the University Hospital of Tübingen (Germany) and digital data are destructed by means of transferring them off memory cards of deployed devices.

The progress of the study is monitored by the principal investigator and other active researchers responsible for carrying out the measurements. The safety standards of deployed devices and equipment are maintained.

## Discussion

This study describes a protocol to primarily investigate the reliability of selected systemic serum biomarkers of inflammation (i.e., CRP, TNF-α, IL-6, IL-1β) by repeated measures over time, among two groups of workers exposed to a higher and lower real-life physical workload, respectively. The findings of this study will be subject to some limitations but may also stand out due to some strengths.

### Study limitations

First, the rat models based on which the current study is designed have presented findings resulting from highly isolated physical exposures [10,11]. In this study, the exposure is not that isolated as in the rat studies and can only be *estimated* in its entirety based on observation using physical workload assessments.

Second, this study will primarily assess the reliability of initially only a few systemic inflammatory biomarkers over time, i.e., CRP, TNF-α, IL-6 and IL-1β, and exploratory investigate their association with musculoskeletal pain. These are just a selection of a large pool of inflammatory biomarkers, of which the temporal stability (i.e., test-retest reliability) of the most suitable inflammatory biomarkers has to be demonstrated. One study has been identified that investigated a large pool of inflammatory biomarkers in women with and without work-related neck/shoulder pain [14]. The authors reported statistically significantly increased levels of CRP in subjects suffering from musculoskeletal neck/shoulder pain (0.5 mg/L, 0.5–1.6) compared to subjects not suffering from musculoskeletal pain (0.5 mg/L, 0.5–0.5) and also reported a statistically significant correlation between CRP and neck/shoulder pain intensity (Spearman ρ 0.43).

Third, the secondary objectives exploratory investigate whether level of musculoskeletal pain and the level of workload exposure may be associated with the level of several systematic inflammatory biomarkers. These objectives should be approached carefully and future longitudinal studies are needed to verify causality.

Fourth, the field study proposed here involves an ambitious study design including nine repeated measures in two groups characterised by a higher and a lower physical workload on the elbow/forearm/hand-area, respectively. Additionally, we aim for twenty participants enrolled in each group with an equal ratio of men and women. These study requirements result in the recruitment of potential participants being challenging. Consequently, we may ultimately have to deal with a possibly sub-optimal ratio within and between both groups of participants regarding sex and potentially also age, BMI and smoking status.

### Study strengths

First, the knowledge of and models currently describing the role of systemic inflammatory biomarkers is based on highly controlled laboratory experiments in rats [10,11]. The current study will be one of the first to attempt to investigate a selection of inflammatory biomarkers in a human population. On top, this human population is examined while working under real-life conditions. This has the advantage that the practical implications for our current explanatory models become much greater due to the higher external validity.

Second, this study aims to determine the temporal stability of a selection of inflammatory biomarkers within a group of people having a constant work-related physical exposure during their daily occupations. We focus on the longer-term

stability of the biomarkers, measuring the participants (as far as possible) before their work shift to determine the reliability of the baseline level of the selected biomarkers over time.

## Supporting information

**S1 Table. SPIRIT-Outcomes 2022 Checklist.**
(PDF)

**S2 File. Ethics review: Study protocol LINOS.**
(PDF)

**S3 File. Translated ethics approval: LINOS.**
(PDF)

## Acknowledgments

The authors would like to thank Esther Herath, head of the outpatient clinic of the Institute of Occupational and Social Medicine and Health Services Research (University Hospital of Tübingen, Germany), for her support in interpreting the differential blood analyses in the screening process and advising us on study enrollment of the screened persons.

## Author contributions

**Conceptualization:** Tessy Luger, Felix Uhlemann, Florestan Wagenblast, Thomas Läubli, Barbara Munz, Manfred Schmolz, Monika A. Rieger, Benjamin Steinhilber.

**Methodology:** Tessy Luger, Felix Uhlemann, Florestan Wagenblast, Thomas Läubli, Barbara Munz, Manfred Schmolz, Monika A. Rieger, Benjamin Steinhilber.

**Supervision:** Tessy Luger.

**Writing – original draft:** Tessy Luger, Felix Uhlemann, Florestan Wagenblast.

**Writing – review & editing:** Tessy Luger, Felix Uhlemann, Florestan Wagenblast, Thomas Läubli, Barbara Munz, Manfred Schmolz, Monika A. Rieger, Benjamin Steinhilber.

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
