## [Decision Letter · Decision Letter 0]

22 Aug 2025

Dear Dr. Luger,

Thank you for submitting your manuscript to PLOS ONE. After careful consideration, we feel that it has merit but does not fully meet PLOS ONE’s publication criteria as it currently stands. Therefore, we invite you to submit a revised version of the manuscript that addresses the points raised during the review process.

I thank the reviewer for the thorough review and constructive comments. A minor revision is required.

We look forward to receiving your revised manuscript.

Kind regards,

Sıdıka Bulduk, Prof. Dr.

Academic Editor

PLOS ONE

Journal Requirements:

“Author MS declares to have a financial competing interest, since he is the CEO and CSO of HOT Screen GmbH (Aspenhaustraße 25, D-72770 Reutlingen, Germany). The remaining authors TLu, FU, FW, TLa, BM, MAR and BS declare that they have no competing interests.”

We note that one or more of the authors are employed by a commercial company: HOT Screen GmbH

Reviewers' comments:

Reviewer's Responses to Questions

**Comments to the Author**

1. Does the manuscript provide a valid rationale for the proposed study, with clearly identified and justified research questions?

Reviewer #1: Yes

2. Is the protocol technically sound and planned in a manner that will lead to a meaningful outcome and allow testing the stated hypotheses?

Reviewer #1: Yes

3. Is the methodology feasible and described in sufficient detail to allow the work to be replicable?

Reviewer #1: Yes

4. Have the authors described where all data underlying the findings will be made available when the study is complete?

Reviewer #1: Yes

5. Is the manuscript presented in an intelligible fashion and written in standard English?

Reviewer #1: Yes

You may also provide optional suggestions and comments to authors that they might find helpful in planning their study.

Reviewer #1: Alcohol consumption may affect the results of inflammatory markers in participants' blood. Could alcohol consumption be an exclusion criterion for the study?.

The location of the algometer's measuring tip: What explains the medial epicondyle?.

**Do you want your identity to be public for this peer review?** For information about this choice, including consent withdrawal, please see our Privacy Policy

Reviewer #1: No

---

## [Author Response · Author response to Decision Letter 1]

19 Sep 2025

Please find our response to the reviewer and editor comments attached as "Response to Reviewers"

---

## [Decision Letter · Decision Letter 1]

24 Sep 2025

Reliability of blood inflammatory markers at constant real-life workloads over time: Study protocol

PONE-D-25-39202R1

Dear Dr. Luger,

We’re pleased to inform you that your manuscript has been judged scientifically suitable for publication and will be formally accepted for publication once it meets all outstanding technical requirements.

Kind regards,

Sıdıka Bulduk, Prof. Dr.

Academic Editor

PLOS ONE

Additional Editor Comments (optional):

Reviewer #1:

Reviewers' comments:

Reviewer's Responses to Questions

**Comments to the Author**

1. Does the manuscript provide a valid rationale for the proposed study, with clearly identified and justified research questions?

Reviewer #1: Yes

2. Is the protocol technically sound and planned in a manner that will lead to a meaningful outcome and allow testing the stated hypotheses?

Reviewer #1: Yes

3. Is the methodology feasible and described in sufficient detail to allow the work to be replicable?

Reviewer #1: Yes

4. Have the authors described where all data underlying the findings will be made available when the study is complete?

Reviewer #1: Yes

5. Is the manuscript presented in an intelligible fashion and written in standard English?

Reviewer #1: Yes

You may also provide optional suggestions and comments to authors that they might find helpful in planning their study.

Reviewer #1: All the necessary corrections have been made to the manuscript. A nice study protocol

I believe it will contribute to the existing literature.

**Do you want your identity to be public for this peer review?** For information about this choice, including consent withdrawal, please see our Privacy Policy

Reviewer #1: No

---

## [Editor Report · Acceptance letter]

PONE-D-25-39202R1

PLOS ONE

Dear Dr. Luger,

I'm pleased to inform you that your manuscript has been deemed suitable for publication in PLOS ONE. Congratulations! Your manuscript is now being handed over to our production team.

Kind regards,

on behalf of

Dr. Sıdıka Bulduk

Academic Editor

PLOS ONE